# Microplastic-Mediated Transfer of Tetracycline Resistance: Unveiling the Role of Mussels in Marine Ecosystems

**DOI:** 10.3390/antibiotics13080727

**Published:** 2024-08-02

**Authors:** Giovanni Milani, Claudia Cortimiglia, Mireya Viviana Belloso Daza, Emanuele Greco, Daniela Bassi, Pier Sandro Cocconcelli

**Affiliations:** Dipartimento di Scienze e Tecnologie Alimentari per una Filiera Agro-Alimentare Sostenibile (DISTAS), Università Cattolica del Sacro Cuore, 29122 Piacenza, Italy; giovanni.milani@unicatt.it (G.M.); claudia.cortimiglia@unicatt.it (C.C.); mireyaviviana.bellosodaza@unicatt.it (M.V.B.D.); emanuele.greco@unicatt.it (E.G.); daniela.bassi@unicatt.it (D.B.)

**Keywords:** microplastics, aquatic ecosystems, biofilm, horizontal gene transfer, antimicrobial resistance

## Abstract

The global threat of antimicrobial resistance (AMR) is exacerbated by the mobilization of antimicrobial resistance genes (ARGs) occurring in different environmental niches, including seawater. Marine environments serve as reservoirs for resistant bacteria and ARGs, further complicated by the ubiquity of microplastics (MPs). MPs can adsorb pollutants and promote bacterial biofilm formation, creating conditions favorable to the dissemination of ARGs. This study explores the dynamics of ARG transfer in the marine bivalve *Mytilus galloprovincialis* within a seawater model, focusing on the influence of polyethylene MPs on the mobilization of the *Tn916*-carrying *tetM* gene and plasmid-encoded *ermB*. Experiments revealed that biofilm formation on MPs by *Enterococcus faecium* and *Listeria monocytogenes* facilitated the transfer of the *tetM* resistance gene, but not the *ermB* gene. Furthermore, the presence of MPs significantly increased the conjugation frequency of *tetM* within mussels, indicating that MPs enhance the potential for ARG mobilization in marine environments. These findings highlight the role of MPs and marine organisms in ARG spread, underscoring the ecological and public health implications.

## 1. Introduction

The emergence of antimicrobial resistance (AMR) is a global threat to human and animal health, occurring in all environments where microbial communities are exposed to anthropogenic use of antimicrobials, such as clinical and farm animal settings [1,2]. Indeed, the excessive and inappropriate use of antimicrobial compounds has led to the escalation of AMR, amplifying its propagation across diverse microbial ecosystems including open environments and food systems [3]. The primary cause for the dissemination of antimicrobial resistance genes (ARGs) is horizontal gene transfer (HGT) via mobile genetic elements (MGEs), such as plasmids and transposons [4]. In particular, conjugative transposons are mobile genetic elements that often carry ARGs and/or virulence genes. These elements play a role in bacterial evolution by imparting particular phenotypes to host cells [5]. HGT may occur in complex matrices like environments, animal guts, and food, where high-density bacterial populations contribute to its occurrence [6,7]. Moreover, in these environments, non-pathogenic and commensal AMR bacteria act as reservoirs of ARGs, which may be transferred to pathogenic bacteria, limiting the effectiveness of commonly used antimicrobial drugs and making infections more difficult to treat [3].

Among the various environmental niches, seawater was shown to act as a reservoir of resistant bacteria originating from human and animal sources, potentially contributing to the dissemination of ARGs. Marine environments were found to enclose about 28% of ARGs [8]. In recent years, the AMR issue has been connected to another major problem for human health, represented by the contamination by microplastics (MPs). MP pollution is increasingly an interdisciplinary field of research, which should be faced with a holistic approach [9]. Defined as plastic fragments with a dimension between 1 μm and 5 mm, MPs have been reported as ubiquitous contaminants of abiotic environments, namely, terrestrial and aquatic environments, and biotic habitats, namely, plants, animals, and humans [9]. This pollution issue constitutes a significant environmental challenge that is difficult to reverse, due to the increasing production, limited recycling, and low degradability of this material. Due to their interactions with chemicals, including antibiotics [10] and microorganisms [11] in the environment, as well as their inherent physicochemical qualities, MPs can impact the overall health and quality of the ecosystem. Indeed, the surface of MP particles is an ideal platform to promote the adhesion of bacteria and the formation of biofilm, creating the so-called ‘plastisphere’. In this condition, microorganisms are protected by a barrier that lets them grow and spread in new and inhospitable settings.

The global dimension of the MP challenge is also due to their movement from urban areas to freshwater and saltwater watersheds. A few years ago, a study estimated the “global annual input of plastic from rivers into the oceans ranging from 1.15 to 2.41 million tonnes with a dominant contribution from rivers of the Asian continent” [12]. Moreover, many studies acknowledged the presence of MPs in wild populations of marine animals [13,14], highlighting possible adverse effects for the animals themselves but also for humans as a source of exposure. Mussels represent a prime example of the bioaccumulation of plastics. Feeding on phytoplankton from surrounding water, they act as a filter, readily also accumulating plastic particles. Therefore, they are used to investigate the fate and the toxic effects of MPs, also recognizing their role in carrying MPs into the food chain [15]. Moreover, the filter-feeding nature of bivalves, combined with human fecal pollution in coastal waters, is suggested to contribute to the bioaccumulation of antibiotic-resistant pathogens in these organisms [16].

Several non-pathogenic foodborne bacteria, such as *Enterococcus faecium*, have raised public health concerns for representing a reservoir of AMR determinants, contributing to their spread to humans through the food chain, animals, and the environment. *E. faecium* is remarkably prone to develop AMR compared to other bacteria, with particular attention to tetracycline, one of the most widely used antimicrobials for treating bacterial infections in humans and animals [17,18]. Moreover, it has been reported that tetracycline resistance in *E. faecium* can be acquired and transferred through transposons, especially the conjugative transposon *Tn916* [19]. In a previous study, we reported described the multi-drug-resistant (MDR) *E. faecium* UC7251, a strain isolated from fermented meat products, highlighting the phenotypic resistance and the genomic organization of genes responsible for these traits. Specifically, this strain carried a mobilizable megaplasmid harboring determinants for resistance to heavy metals and to numerous antimicrobials including aminoglycosides (*ant(6)-Ia*, *aph(3′)-III*, *aad(6)-Ia*, *ant1*), macrolides (*ermB*, *mrsC*, *satA*), lincosamides (*IsaE*, *LnuB*), and tetracyclines (*tetL*), embedded in an integrative and conjugative element (ICE). In addition, *Tn916* carrying the tetracycline resistance gene *tetM* was detected on the chromosome. A conjugation experiment between *E. faecium* UC7251 and other strains showed the successful in vitro transfer of only the chromosomal *tetM* gene of this determinant from this strain, isolated from fermented meat to *Listeria monocytogenes* [20]. The latter, an important foodborne pathogen frequently detected in cheese, meat, and seafood products, has been shown to be able to acquire or transfer AMR genes. The ability to acquire genetic material from other microorganisms has been observed in *L. monocytogenes*, which is recognized as an important food pathogen, frequently detected in cheese, meat, and fish products [21,22].

Several studies have investigated the bacterial community and the antimicrobial resistome on MPs, providing useful insights on the prevalence of ARGs and the variety of microbial species accumulating in the plastisphere. Although recent research suggested links between the presence of MPs and ARGs in seawater and fresh water, research into the demonstration of these links is still lacking. Since currently there are no studies that experimentally demonstrate the horizontal gene exchange in seafood in the presence of MPs, we studied the transfer dynamics using the bivalve *Mytilus galloprovincialis* in a seawater model, examining the influence of polyethylene MPs on HGT.

## 2. Results

### 2.1. Involvement of MPs in ARG Exchange inside Mussels

The aim of this study is to evaluate the role played by MPs and mussels in transferring AMR in the marine environment, and then to assess how the combination of both elements affects the passage of mobile elements. In particular, the exchange of the determinants responsible for tetracycline carried by the conjugative transposon Tn916 located on the chromosome and erythromycin, carried by mobilizable plasmid pUC7251_1, between *E. faecium* and *L. monocytogenes*, two species found in the marine environment, was assessed. Firstly, we tested the ability of donor and recipient to form biofilms on plastic material. According to the crystal violet test, the ability of *E. faecium* UC7251 and the two strains of L. monocytogenes to produce biofilm was confirmed, finding OD595 of 0.488, 0.448, and 0.420 for *E. faecium* and *L. monocytogenes* DSM 15675 and Scott A, respectively. Both enterococci and *L. monocytogenes* are known to produce biofilm, as observed by others [23,24,25]. Then, we proceeded to set up the experimental conditions, adding biofilm-covered MPs and mussels alone or in combination, and also checking whether HGT could occur between planktonic cells. When *E. faecium* and *L. monocytogenes* planktonic cells were added in seawater, in the presence or absence of mussels, we did not observe any conjugation events concerning the *tetM* and *ermB* genes, even if both strains persisted in the environment.

The HGT of *tetM* gene, but not of the *ermB* gene, was detected when MPs individually covered by donor and recipient were added to the aquarium (experimental condition ii) (Appendix A). Indeed, the transfer of the *tetM* gene from *E. faecium* to *L. monocytogenes* DSM 15675 occurred with a rate of −7.91 ± 0.12 Log T/D (Transconjugants/Donor) after four days, increasing significantly at −6.41 ± 0.18 Log T/D after seven days (Figure 1A). The *tetM* transfer frequency observed for *L. monocytogenes* Scott A was not statistically different from that observed for *L. monocytogenes* DSM 15675 (Appendix A).

Furthermore, we investigated the ability of *M. galloprovincialis*, a filter-feeding aquatic organism, to act as a host of HGT between bacterial communities. When MPs covered by a biofilm of *E. faecium* and *L. monocytogenes* DSM 15675 were added to the marine environment, the analysis of mussels resulted in a *tetM* conjugation rate of −5.48 ± 0.07 Log T/D after 4 days (Appendix A). Statistically not different transconjugation values were detected when the recipient strain was *L. monocytogenes* Scott A (Figure 2B). On the seventh day, a statistical increase in conjugation rate was achieved for both recipient strains, without difference between them. In this experimental condition, water samples were also analyzed, finding a T/D ratio of −6.91 ± 0.05 Log T/D and −6.83 ± 0.06 Log T/D after 96 h, and a T/D ratio of –6.60 ± 0.04 Log T/D and −6.68 ± 0.10 Log T/D after 168 h for *L. monocytogenes* DSM and Scott A, respectively (Figure 1B, Appendix A). Also in this experimental condition, we did not observe any *L. monocytogenes* colonies on ALOA with erythromycin, and therefore no ermB gene transfer events occurred.

No colonies referable to Listeria genus were detected on ALOA supplemented with tetracycline or erythromycin in negative control.

### 2.2. The Role of MGE of L. monocytogenes in HGT

The absence of any statistically significant differences between conjugation rates obtained with *L. monocytogenes* DSM and Scott A strains led us to check the possible involvement of MGE transfer.

The genome investigation for MGEs in *L. monocytogenes* DSM 15675 strain resulted in the absence of any type of MGE, whereas *L. monocytogenes* Scott A presented an integrative conjugative element of 58 kb (2375271-2433301bp) containing the conjugative transposon Tn5422 which does not carry AMR genes. These results were consistent with previous works which demonstrated that this is primarily attributed to the existence of a Type IV secretion system (T4SS) and a relaxase in donor cells that promotes the transfer of genes to the recipient cell without requiring these proteins to be present in the recipient cell itself [26].

## 3. Discussion

Investigating the sharing of ARGs within a microbial niche is a priority, especially if it involves ecological environments that directly impact on human health [1]. Aquatic environments have been shown to be vehicles of pathogenic microorganisms and ARGs, along with other pollutants such as MPs. Given the complexity of the ecological system in which we currently live, it is particularly important to investigate what events drive gene transfer.

Here, we investigated the ability of an MDR *E. faecium* to share the *tetM* and *ermB* determinants with two *L. monocytogenes* strains in an artificial seawater environment, including MPs and mussels. First of all, we observed that even though the planktonic cells were not able to exchange *tetM* and *ermB*, both of them persisted in this situation, accordingly with some studies which detected the presence of these microorganisms in the marine ecosystem [27,28,29,30].

Moreover, our study demonstrated that the HGT of *tetM,* but not *ermB*, occurred only when the MPs were covered by donor and recipient biofilm, and that the conjugation rate was increased when the filter feeding mussel *M. galloprovincialis* was present in the same artificial environment. As supposed by others, probably the high cellular density and the protective action of biofilm allowed the *Tn916*-mediated conjugation to occur, albeit with low frequency and despite the absence of any selective pressure, such as the presence of antibiotics [31,32]. Indeed, the biofilm is believed to serve as a focal point to promote the mobilization of ARGs, also in aquatic environments, where it was detected on different types of surfaces, namely, rocks and plastics [33]. The aggregation in biofilms lead bacteria to be protected from different types of stresses such as temperature, UV, salinity, and pH changes [34], thus representing a strategy for bacteria to remain in this challenging ecosystem. Despite favorable conditions in the biofilm that increase contact between donor and recipient, the absence of *ermB* gene transfer could be due to the absence of supporting structures (e.g., helper plasmids) in the donor strain or to the absence of selective pressure from subinhibitory concentrations of the antibiotic during experiments [35].

Although some studies observed that ARGs persisted in different types of aquatic environments such as ocean and marine sediments, leading to the hypothesis that they are capable of being spread in bacterial communities, few studies directly demonstrated the HGT mediated by MPs in marine ecosystems. Arias-Andres and coauthors evidenced conjugation events in an experimental microcosm study, in the presence or absence of MPs. They observed the colonization of MPs by donor and recipient strains and an increase of 4 LOG in the conjugation rate when MPs were included [36]. In another study, the MP biofilm was demonstrated to increase the natural transformation rate of extracellular DNA in single cells and the same-niche biota, unlike when only planktonic cells were present [31].

Since the mussel *M. galloprovincialis* is a valuable edible marine bivalve with significant ecological and economic importance, the evaluation of its role in the mobilization of ARGs has become of interest. Previous studies have assessed the presence of pathogenic and AMR bacteria in bivalve mussels and seafood, revealing a problem that needs to be monitored. Surprisingly, no HGT events were found without MPs, highlighting that both donor and recipient strains were able to persist in seawater without finding proper conditions inside mussels to exchange *tetM* or *ermB* genes. The mobilization of ARGs could probably be fostered in the presence of antibiotics, since they are frequently detected in different aquaculture facilities and in marine ecosystems [37,38,39]. However, considering the experimental conditions with MPs, the analysis of *tetM* transfer in water and within mussels showed that the frequency of conjugation occurring in mussels is statistically higher (*p* < 0.05) than that found in water. This evidence could be due to the fact that *M. galloprovincialis*, as a filter feeder, is able to accumulate plastics inside the gut, according to the literature. Indeed, mussels have long been recognized as indicators of plastic pollution because of many different characteristics, such as their global distribution, easy accessibility, and high tolerance to environmental conditions. Monitoring activities over the years have shown the presence of MPs in the gut of different species of mussels, observing a positive correlation between the abundance of MPs in water and in the mussels, independently from the types of MPs (i.e., beads, fragments, and fibers) [15,40,41]. Moreover, higher levels of MP ingestions were demonstrated in mussels compared to other fishes, such as pandoras (*Pagellus erythrinus*) and red mullets (*Mullus barbatus*) [41].

The filtering action of MPs exerted by mussels is combined with the presence of biofilms on them, creating even more favorable conditions for gene exchange. Indeed, several studies have found the presence of biofilms on MPs [42,43], and that biofilms increase gene exchange frequencies [36,44]. Therefore, we overall hypothesized that mussels are biological amplifiers of HGT, as they exert filtering action and concentrate MPs covered by biofilm.

## 4. Materials and Methods

### 4.1. Bacterial Strains and Culture Conditions

*E. faecium* UC7251 was used as the donor strain and cultured in Brain Heart Infusion (BHI) (Oxoid, Cheshire, UK) supplemented with 10 µg/mL of tetracycline (Sigma Aldrich, Saint Louis, MO, USA) and 50 µg/mL of erythromycin (Sigma Aldrich) and incubated at 37 °C overnight. Two *L. monocytogenes* strains, DSM 15675 and Scott A, cultivated in BHI and incubated at 37 °C overnight, were used as recipient strains.

*E. faecium* (GCA_000411655.2) and *L. monocytogenes* genomes (Scott A: GCA_009866905.1; DSM 15675: GCA_002156185.1) were downloaded from NCBI and screened with ICEfinder [45] to find integrative and conjugative elements (ICEs), including chromosome-borne integrative and mobilizable elements (IMEs) and cis-mobilizable elements (CIMEs).

### 4.2. L. monocytogenes and E. faecium Biofilm Formation: Microtiter Assay and Microplastic Colonization

The biofilm formation ability of *E. faecium* UC7251 and *L. monocytogenes* DSM 15675 and Scott A was quantified on 96-well polystyrene micro-titer plates (Sarstedt, Nümbrecht, Germany) in triplicate, as previously described [46]. Biofilm formation was evaluated by measuring the absorbance at OD595 nm using a Multiskan EX (Thermo Electron Corporation, Waltham, MA, USA).

The formation of biofilm on Polyethylene (PE) microplastic by *L. monocytogenes* and *E. faecium* was carried out as described by others with some modifications [46]. Five grams of PE, with a molecular weight of 35,000 g/mol (PE 35000) (Sigma Aldrich) and a particle size ranging from 10 to 2000 µm (mean 271 µm), were pre-sterilized with ethanol 95% for 24 h and placed into glass flasks with 100 mL of BHI. Three glass flasks were inoculated with 100 µL of overnight cultures of *E. faecium* UC7251, *L. monocytogenes* DSM 15675, and *L. monocytogenes* Scott A, respectively, and incubated at 37 °C without agitation for 6 days. After this time, the MPs were recovered aseptically and washed three times with distilled water. Biofilm formation was determined as previously described [46]. After that, MPs individually coated with the donor and recipient biofilm were used in the conjugation experiment explained in the following section.

### 4.3. Experimental Design

The assessment of AMR gene transfer was studied in vivo in mussels with or without MPs in a controlled aquatic environment using artificial seawater ASTM D1141-98 (Thermo Fisher Scientific, Waltham, MA, USA) in a 60 L fish tank with water recycling and oxygenation. The *tetM* and *ermB* genes, harbored by chromosomes and plasmids, respectively, were considered to monitor the gene exchange between donor and recipients. The experimental design (Figure 2) included four conditions: (i) artificial seawater inoculated with *L. monocytogenes* and *E. faecium* as planktonic cells; (ii) seawater with MPs covered by *L. monocytogenes* and *E. faecium* biofilms (without mussels); (iii) mussels (*M. galloprovincialis*) in artificial seawater inoculated with donor and recipient as planktonic cells; and (iv) mussels in seawater with MPs individually coated with *L. monocytogenes* and *E. faecium* biofilms.

For setting up conditions (iii) and (iv), 60 undamaged and live adult mussels were added to the fish tank, respectively. In conditions (i) and (iii), overnight cultures of the donor and recipient strains were washed three times with saline solution (0.9% NaCl) and inoculated into the seawater with a final concentration of planktonic cells of 1 × 10^6^ CFU/mL, respectively. For conditions (ii) and (iv), 5 g of MPs coated with donor biofilm (load of *E. faecium*: 6 Log CFU/mL) and 5 g of MPs coated with recipient biofilm (load of *L. monocytogenes* DSM 15,675 and Scott A: 6 Log CFU/mL) were used. The temperature in the aquarium was maintained at 15 °C for seven days. An aquarium with mussels in marine water, without inoculum or MPs, was used as negative control.

### 4.4. Microbiological and Molecular Analysis for Transconjugants Confirmation

After 4 and 7 days, mussels and water were sampled and analyzed to assess the frequency of transconjugants.

Three replicates of 10 mL of seawater were mixed with 90 mL of saline solution and homogenized with stomacher three times for five minutes. For mussels, three replicates of 10 bivalves were separated from the shell, diluted, and homogenized as described above. Samples were serially diluted and plated on ALOA (Oxoid) supplemented with 10 µg/mL of tetracycline or 50 µg/mL of erythromycin to select and count transconjugants, and on Slanetz and Bartley agar (Oxoid) supplemented with tetracycline (10 µg/mL) and erythromycin (50 µg/mL) to quantify the donor strains. The conjugation frequencies were calculated as the ratio between the concentration of the transconjugants and the concentration of the donor strain [47].

The transconjugants were confirmed using a PCR assay targeting the *tetM* and *ermB* genes, as previously described [20], and species-specific PCR targeting *actA* gene for *L. monocytogenes* [48].

### 4.5. Statistical Analysis

Statistical analysis was performed using the Past 4.06b software package [49]. Conjugation frequencies were analyzed using one-way analysis of variance (ANOVA) with Tukey’s multiple comparison test (*p* ≤ 0.05). Experiments were conducted in a BL2 (Biosafety Level 2) Bacteriology Laboratory, in accordance with the WHO guidelines [50].

## 5. Conclusions

In conclusion, this study highlights the significant role of MPs and marine organisms in mediating the horizontal transfer of ARGs among strains of *E. faecium* and *L. monocytogenes*, frequently found in the marine environment. The findings underscore the potential ecological and public health implications associated with the dissemination of AMR in aquatic environments.

Our investigation demonstrated that the presence of MPs, particularly when coated with bacterial biofilms, significantly enhanced the HGT of the *tetM* gene between *E. faecium* and *L. monocytogenes* strains. This phenomenon suggests that MPs may serve as platforms that concentrate bacterial cells and promote genetic exchange, potentially accelerating the spread of AMR. Moreover, the study highlights the role of filter-feeding organisms like *M. galloprovincialis* in facilitating HGT within aquatic environments. Mussels ingesting MPs coated with bacterial biofilms demonstrated increased rates of gene transfer, emphasizing the potential of marine organisms to act as vectors for AMR dissemination. The accumulation of MPs in mussels has also an impact on food safety. Indeed, their ingestion represents a potential risk for human health and a reservoir of AMR determinants.

In summary, this study provides crucial insights into the complex interactions between MPs, marine organisms, and bacterial communities in driving the spread of AMR in aquatic ecosystems. Further research is warranted to elucidate the broader ecological implications of AMR dissemination facilitated by MPs and to develop strategies for mitigating this emerging global health threat.

## Figures and Tables

**Figure 1 antibiotics-13-00727-f001:**
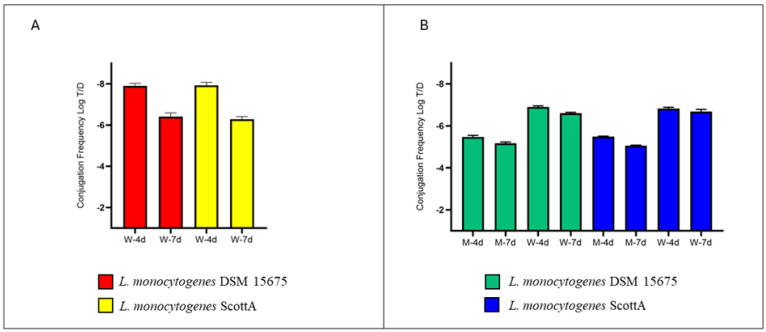
Conjugation rates (T/D) found in two experimental conditions: (**A**) seawater with MPs covered by *L. monocytogenes* and *E. faecium* biofilms (without mussels); (**B**) mussels in seawater with MPs individually coated with *L. monocytogenes* and *E. faecium* biofilms. The x axis reports the name of samples (M: mussels; W = water) and time of sampling (4d: fourth day; 7d: seventh day).

**Figure 2 antibiotics-13-00727-f002:**
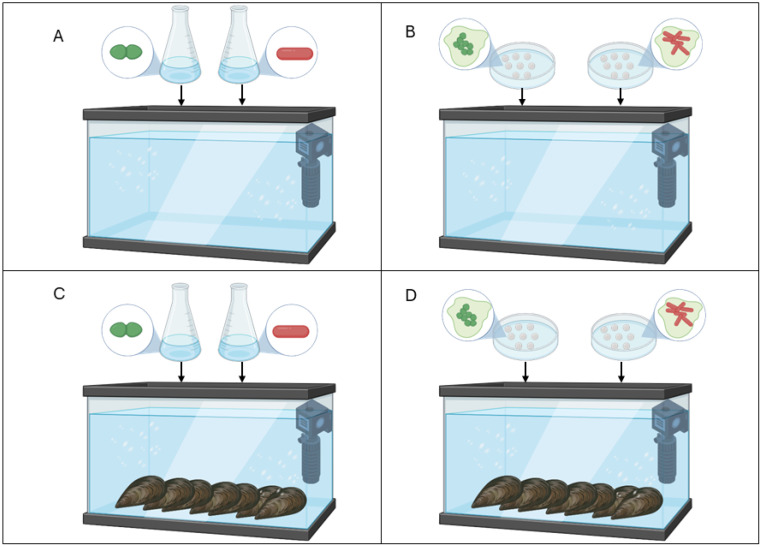
Experimental design, comprising four conditions: (**A**) experimental condition (i) with planktonic cells of *E. faecium* UC7251 and *L. monocytogenes* strains; (**B**) experimental condition (ii) including MPs covered by a biofilm of donor and recipients, prepared separately and simultaneously inoculated; (**C**) experimental condition (iii) including an artificial marine environment with mussels and water inoculated with planktonic cells of donor and recipients; and (**D**) experimental condition (iv) containing mussels and MPs covered by donor and recipient biofilm.

## Data Availability

Data are contained within the article and Appendix A.

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
