# Peer review of "Microplastic-Mediated Transfer of Tetracycline Resistance: Unveiling the Role of Mussels in Marine Ecosystems"

_antibiotics, 2024, doi:10.3390/antibiotics13080727_

Round 1

Reviewer 1 Report

Comments and Suggestions for Authors

The authors conduced an interesting experiment in which they observed the facilitation of the transfer of the tetracycline antibiotic resistance gene (tetM) mediated by microplastics. The results of the experiment in a marine model using the mussel M. galloprovincialis showed that the rate of gene transfer increases when bivalves are coated with microplastics containing donor and recipient straing, suggesting the role of these organisms as vector for the dissemination of antimicrobial-resistant microorganisms. The authors highlight the implications for food safety and public healt that the possibility of ingesting mussels with microplastics that promote the formation of biofilms entails.

Minimal observations:

Add reference after:

E. faecium is remarkable prone to develop.............infections in humans and animals.

Add reference after:

Furthermore, it has been reported that tetracycline resistance..............conjugative transposon Tn916.

Delete the letter "A" in the la sentence: "A In a previous study......" row 74

Author Response

Please, see the attachment. We have highlighted the answers in red.

Reviewer 2 Report

Comments and Suggestions for Authors

The manuscript entitled "Microplastic mediated transfer of tetracycline resistance: unveiling the role of mussels in marine ecosystem" is nicely presented and interesting to read. The work designed well without any flaws and the same may be recommended for publication with minor modifications.

1. The MP contamination is the biggest problem that we encounter in current scenario, the  importance and statistics can be included to enrich the highlight of this study. 

2. The role of M. Galloprovincialis in HGT  may be discussed with additional references.

3. A possible hypothesis behid the HGT is provided in line 202 and 203. However, a deeper insight can be given by the authors with additional data from the literature would open new avenues of research in this area.

Author Response

(The authors gave the same response as above.)

Reviewer 3 Report

Comments and Suggestions for Authors

General comments:

This manuscript investigated the horizontal gene transfer of antibiotic resistance genes (ARGs) in marine systems. The study found that microplastics (MPs) and the presence of mussels significantly enhanced the transfer of the tetM gene between E. faecium and L. monocytogenes strains. The article is relevant with growing concerns about MPs and ARGs in the scientific community. It is recommended that the manuscript be accepted for publication after addressing the following comments:

In the method section, the authors described utilizing 5g of microplastics and 1x106 CFU/ml bacteria for the experiments with microplastics and planktonic cells, respectively. Please explain the rationale behind the chosen concentrations.

Specific comments:

·      Line 141: The authors stated that no tetracycline resistance L. monocytogenes was detected in negative control. Please clarify if both plate counts and PCR experiments demonstrated the same results.  

·      Line 250: Please clarify whether 1x106 CFU/ml is the final concentration of the planktonic cells or the inoculant concentration.

·      Figure 2: The figure caption is not consistent with the description in Lines 241-246. Please revise.

Author Response

(The authors gave the same response as above.)

Reviewer 4 Report

Comments and Suggestions for Authors

The article concerns the important issue of antibiotic resistance in the environment, taking into account microplastic particles as vectors promoting the formation of biofilm and the spread of ARGs between microorganisms. The article is concise and valuable. It is not overly extensive in the text, but I do not consider it a disadvantage but a benefit. The conducted research shows clear conclusions. The research design is interesting and the research concept is well organized. However, I think the article has some defects. I recommend that the authors re-read and improve the quality of the manuscript. Perhaps some linguistic improvement of the manuscript would be needed. I present several comments below, that may draw attention to the mentioned weaknesses of the manuscript:

The authors introduced the abbreviation “MGEs” for mobile genetic elements. To be consistent, in lines 34 and 100 (and perhaps also in other parts of the manuscript) the authors should use the introduced abbreviation. Moreover, for example in line 196 – “resistance genes” in text – why not “ARGs” if authors introduced this abbreviation?

Lines 67-88 - the authors explain why the tetM gene was chosen for the research concept. It is not clear why one of the erythromycin resistance genes was also selected for analysis. Moreover, there is no explanation for the selection of the ermB gene (from erm gene class). This requires some explanation. What's more, the authors mention the tetL gene in the methods... (?) Unfortunately, the manuscript is messy.

Line 101 “determinants responsible for tetracycline and erytromycin” - why did authors decide to include the erythromycin resistance? Please explain. Moreover, not "erytromycin", but "erythromycin"

104 “two L. monocytogens” -  the authors mean about two L. monocytogenes strains? There are more such that kind of mistakes in the manuscript.

112 “tetM and ermB genes” – It remains a mystery why this set of genes was selected for analysis. It is very important to justify this decision.

Line 144 “2.2. Mobile genetic elements in L. monocytogenes strains don’t affect the HGT” - I suggest changing the title of the subsection. In this form, it sounds more like a conclusion.

Line 183 “Arias-Andres and colleagues” or line 221 “described by Hchaichi and colleagues” - the term " colleagues" is probably not the most accurate (informal).

Figure 2 should illustrate the design of the experiment (line 255). Unfortunately, the legend is missing, so the figure is not intuitive and understandable enough. Moreover, the quality of Fig. 2 could be better.

Lina 271 “10 μg/ml of tetracycline or 50 μg/ml of erythromycin” - There is no explanation why certain concentrations of antibiotics were chosen and not other ones.

Line 177 - why tetM, tetL, and ermB genes? The authors mention the tetL gene only here... This is completely confusion.

Line 178  there is no information in the text on which species-specific genes were amplified in PCR

Comments on the Quality of English Language

Perhaps some linguistic improvement of the manuscript would be needed.

Author Response

(The authors gave the same response as above.)
